# Sufentanil vs. Dexmedetomidine as Neuraxial Adjuvants in Cesarean Section: A Mono-Centric Retrospective Comparative Study

**DOI:** 10.3390/jcm11226868

**Published:** 2022-11-21

**Authors:** Antonio Coviello, Carmine Iacovazzo, Anella D’Abrunzo, Marilena Ianniello, Maria Grazia Frigo, Annachiara Marra, Pasquale Buonanno, Maria Silvia Barone, Giuseppe Servillo, Maria Vargas

**Affiliations:** 1Department of Neurosciences, Reproductive and Odontostomatological Sciences, University of Naples “Federico II”, 80100 Naples, Italy; 2Department of Anesthesia and Resuscitation in Obstetrics, San Giovanni Calibita Fatebenefratelli Hospital, 39, 00186 Rome, Italy

**Keywords:** caesarean delivery, intrathecal, dexmedetomidine, intrathecal sufentanil, postoperative pain, spinal anesthesia

## Abstract

Spinal anesthesia is the best choice for caesarean delivery. This technique is characterized by a complete and predictable nerve block with a fast onset and few complications. Several intrathecal adjuvants are used in order to improve the quality and duration of anesthesia and reduce its side effects. Sixty-two patients who underwent caesarean delivery under spinal anesthesia were included in this medical records review. In this retrospective study, after adopting exclusion criteria, we assessed 24 patients who received Hyperbaric Bupivacaine 0.5% 10 mg and dexmedetomidine 10 μg (G1), and 28 patients who received an institutional standard treatment with Hyperbaric Bupivacaine 0.5% 10 mg and sufentanil 5 μg (G2). We evaluated the difference in terms of motor and sensory block, postoperative pain, and adverse effects during the first 24 h following delivery and neonatal outcome. Our study found that the sufentanil group had a significantly lower requirement for analgesia than the dexmedetomidine group. Postoperative pain, assessed with the VAS scale, was stronger in G1 than in G2 (4 ± 2 vs. 2 ± 1, *p*-value < 0.01). Differences between the two groups regarding the intraoperative degree of motor and sensory block, motor recovery time, and neonatal Apgar scores were not noticed. Pruritus and shivering were observed only in G2. Itching and shivering did not occur in the dexmedetomidine group. Postoperative analgesia was superior in the sufentanil group, but the incidence of side effects was higher. Adjuvant dexmedetomidine prevented postoperative shivering.

## 1. Introduction

Spinal anesthesia is the ideal choice for elective or urgent caesarean delivery [1]. This technique is associated with a dense and predictable block; therefore, it shows a faster onset and fewer complications compared to other anesthetic protocols [2]. Spinal anesthesia presents adverse reactions that greatly threaten maternal and fetal safety. They include shivering, nausea and vomiting, hypotension, and bradycardia. Shivering can influence maternal metabolic activity and maternal hypotension can provoke a reduction in the blood supply towards the placenta, which can result in hypoxia and acidosis in the fetus [3]. Several studies have proven that by reducing the intrathecal local anesthetic dose, the incidence of adverse events is decreased [4]. However, a lower dosage was associated with a reduced duration of anesthesia and analgesia. Several adjuvants are widely used for intrathecal local anesthetics to enhance the quality and to extend the duration of anesthesia and analgesia by reducing the dose of local anesthetics. Opioids (morphine, fentanyl, and sufentanil) and α2 adrenergic agonists (dexmedetomidine and clonidine) are more commonly used as adjuvants in clinical practice [5]. The supplementation of opioids is linked to side effects such as itching, nausea/vomiting, hypotension, and delayed respiratory depression, which appear to be dependent on the lipophilic character of the opioid used. Among opioids, sufentanil 5 mcg appears to be connected to the best quality of anesthesia and analgesia with no increase in side effects [1]. The literature has shown that the intrathecal use of α2 adrenergic agonists does not cause nausea and vomiting episodes; instead, they prolong analgesia and can decrease the occurrence of shivering during caesarean delivery [3,6,7]. Dexmedetomidine, as an adjuvant, can prolong the duration of analgesia of a local anesthetic in spinal anesthesia [8]. Hemodynamic effects such as bradycardia or hypotension of the α2 adrenergic agonists, as well as their effects on the fetal outcome, depend on the dose [9,10,11]. The use of intrathecal adjuvants, with doses that the literature has shown to be safe, did not affect neonates. In fact, there were no differences in umbilical cord blood gases and neonatal Apgar scores [8,12,13]. In this investigation, our target is to compare the use of intrathecal dexmedetomidine and sufentanil in pregnant women who underwent caesarean delivery in order to evaluate the intraoperative degree of motor block assessed by the Bromage scale and the sensory block level assessed by the Hollmen scale. The neonatal Apgar score was evaluated at birth. In the first 24 h of postoperative pain assessed via the VAS scale, adverse effects such as nausea, vomiting, pruritus or shivering, first flatus time, and motor recovery time were assessed.

## 2. Materials and Methods

This was a Level III monocentric, retrospective, comparative study performed at the Department of Neurosciences, Reproductive and Odontostomatological Sciences, “Federico II” University (Naples, Italy). The study did not require the IRB’s approval because of the retrospective study design; the data were anonymized before analysis. All procedures performed in this study were also in accordance with the 1964 Declaration of Helsinki and its later amendments or comparable ethical standards. The Strengthening the Reporting of Observational Studies in Epidemiology (STROBE) statement was followed.

### 2.1. Inclusion/Exclusion Criteria

The information of patients who underwent caesarean delivery (CS) between June 2021 and December 2021 at our institution was obtained from the archive of the department, recorded on a pre-filled form, and stored in a password-protected, computerized database using MS Office Excel 2007 (Microsoft, Redmond, WA, USA).

Inclusion criteria were as follows: full-term pregnant women with ASA (American Society of Anesthesiologists) physical status of I–II, aged between 18 and 45, and undergoing elective caesarean delivery under spinal anesthesia.

Exclusion criteria were as follows: patients with preeclampsia, ASA of III–IV, a pregnancy term other than 36–40 weeks, multiple gestation, hypersensitivity, and contraindications to spinal block.

### 2.2. Study Population

In the timeframe considered, 62 patients underwent CS. After implementing the exclusion criteria (3 with preeclampsia, 5 pregnancies before 36 weeks, and 2 multiple gestations), 52 patients were included in the study.

### 2.3. Interventions

In the operating theater, a venous access was implemented (18–16 G), and Pantoprazole 40 mg iv, ondansetron 8 mg iv, and antibiotic prophylaxis were administered (Cefazolin 1 or 2 g IV or, in case of allergy, clindamycin 600 mg iv) 30 min before skin incision. Electrocardiogram (ECG), Pulse Oximetry (SpO2), Body Temperature (TC), and Continuous Non-Invasive Blood Pressure (NIBP) every 2.5 min until fetal extraction—and then every 5 min—were monitored. A co-loading was initiated with 500 mL of IV crystalloids. We proceeded to perform an intraoperative fluid administration of 15–20 mL/kg/hour of IV crystalloids. Spinal anesthesia was performed at the L2–L3 or L3–L4 interspace with the patient in the left side position. Vertebral level was recognized starting from the sacrum. We counted the laminae in the caudal-to-cephalad direction, which was then marked with a surgical pen. The technique was performed aseptically in the subarachnoid space after observing clear Cephalo-Spinal Fluid (CSF) in the spinal needle 27 Gauge, and without releasing the CSF. Selected patients received Hyperbaric Bupivacaine 0.5% 10 mg and dexmedetomidine 10 μg or Sufentanil 5 μg. The position of the uterus was secured to the left until birth by inserting a Crawford wedge after placing patients in supine position. In case of maternal hypotension (MAP <60 mmHg, SAP <90 mmHg, or <20% of the initial values), Ephedrine 0.1 mg/kg IV was administered if it was associated with maternal bradycardia (Heart rate < 60 bpm); alternatively, IV Phenylephrine 100 mcg/each bolus was administered (Appendix A) if maternal hypotension was associated with tachycardia (Heart rate > 100 bpm). At fetal extraction time, IV Oxytocin 5 IU IV was administered as a slow bolus (3 min for all patients and 5 min in case of cardiopathic patients). This was followed by 10 IU/hour of Oxytocin at 500 mL in slow infusion in the following post-partum 2/4 h. Soon after birth, Oxytocin infusion was initiated. In case of poor uterine contraction or postpartum hemorrhage (defined as blood loss exceeding 1000 mL), methylergometrine 0.4 mg was IM administered. Operation started when block reached the T5 level. In the postoperative period, VAS assessment was carried out with a 10 cm long line with verbal anchors at either end (“no pain” on the far left and “the most intense pain” on the far right). The patient marked a point on the line corresponding to the rating of pain intensity. After surgery, we administered intravenous paracetamol 1 g 3 times a day. Keterolac 30 mg was available as rescue dose. Pain control was considered good in case of VAS score less than 4. The patients were evaluated by clinicians every 6 h in each postoperative period (first 24 h) to determine: VAS; presence of adverse effects such as nausea, vomiting, pruritus, or shivering; first flatus time; and motor recovery time. Pharmacological therapy was based on the patients’ responses. Presence of nausea, vomiting, or shivering was treated with 8 mg of ondansetron.

### 2.4. Data Extraction

The following data were obtained from medical records: age, BMI, ASA physical status, gestational age, previous pregnancy, morbidity during pregnancy, and caesarean delivery indication. Furthermore, surgery time, incidence of hypotension, and intraoperative blood loss were evaluated. Intraoperative degree of motor block measured with Bromage scale and sensory block level were evaluated with Hollmen scale. The neonatal Apgar was evaluated at birth (Appendix A). Postoperative pain with VAS scale; presence of adverse effects such as nausea, vomiting, pruritus, or shivering; first flatus time; and motor recovery time were assessed in the first 24 h.

### 2.5. Statistical Analysis

Patients were divided into Group 1 and Group 2:Group 1 (G1): Bupivacaine 0.5% 10 mg and dexmedetomidine 10 μg;Group 2 (G2): Bupivacaine 0.5% 10 mg and Sufentanil 5 μg.

Categorical variables were reported in percentage and compared with chi-square test. Continuous variables were reported as mean and standard deviation and compared with the Student’s *t*-test for unpaired samples. Statistical significance was set at ***p*** 0.05. Statistical analysis was performed using the IBM SPSS software (version 20.0, IBM Corporation, New York, NY, USA).

## 3. Results

### 3.1. Demographic Data

In the selected cohort, 24 patients (46%) were included in G1 and 28 patients (54%) in G2. No relevant difference in the demographic data between the two groups was observed, even with respect to one or more previous pregnancies as reported in Table 1. No patient included in the study had a weight gain greater than 14 kg during pregnancy. Only one pregnant woman, in the sufentanil group, suffered from gestational diabetes. All patients had a physiological pregnancy; three patients from the dexmedetomidine group were at risk for preterm delivery and were forced to rest during the gestational period. No patient showed anxiety in their medical history.

### 3.2. Intraoperative Degree of Motor Block and Sensory Block Level

No difference regarding the onset of sensory block and the intraoperative degree of motor block of the two groups evaluated with the Bromage scale and sensory block level evaluated with the Hollmen scale was observed. The motor recovery time was not significantly different between the two groups.

### 3.3. Surgical Data

The caesarean delivery indications are reported in Table 2. The duration of surgery between the two groups is similar. Surgical complications or major blood loss were not observed for any patient. The incidence of hypotension is not statistically significant between the two groups; data are shown in Table 1.

### 3.4. Postoperative Pain

The levels of postoperative pain in the first 24 h assessed via the VAS scale were higher in G1 than G2 (4 ± 2 vs. 2 ± 1, *p*-value < 0.01).

### 3.5. Adverse Effects

#### 3.5.1. Shivering

Shivering was observed only in G2. The results showed that the difference was not statistically significant (*p* > 0.05).

#### 3.5.2. First Flatus Time

The first flatus time was lower in G1 compared to G2 (within 12 h, 100% for G1 vs. 25% for G2, *p* < 0.05).

#### 3.5.3. Nausea, Vomiting, and Itching

The incidence of nausea and vomiting was higher in G2 compared to G1, but the difference was not statistically significant (*p* > 0.05). Itching was observed in the group receiving sufentanil; on the contrary, intrathecal dexmedetomidine did not affect the occurrence of itching.

#### 3.5.4. Neonatal Apgar

There were no differences in the neonatal Apgar scores at 1 and 5 *min* between the two groups (Table 3).

## 4. Discussion

In this study, spinal anesthesia carried out with fine gauge disposable needles and the introduction of bupivacaine along with adjuvant drugs into the subarachnoid space was proven to be an incisive technique for a caesarean delivery. In fact, the intrathecal administration of bupivacaine with dexmedetomidine or sufentanil determined an adequate anesthetic plane with complete motor block (Bromage scale 1) and sensory-level block tested via a pinprick and ice tests at least at the T5 level. A superior degree of analgesia in the first postoperative 24 h was achieved for the pregnant women that received intrathecal sufentanil compared to the patients that received dexmedetomidine as an intrathecal adjuvant. In the pregnant women that received intrathecal sufentanil, itching was the most important side effect that our study reported, which is in accordance with the analysis of previous studies. There was not a substantial difference between the two groups concerning the other side effects evaluated in this study. As in previous studies [12,13,14,15], this study also did not find any differences in the neonatal Apgar scores among the groups administered dexmedetomidine or sufentanil. It is not clear if any adverse neonatal effects related to the use of the intrathecal adjuvant occurred.

Nowadays, spinal anesthesia with a T4–T6 level is the most common form of regional anesthesia performed for caesarean delivery. Spinal anesthesia is useful for an urgent caesarean delivery when a rapid induction of anesthesia is required. The selection of a local anesthetic for spinal anesthesia is based primarily on the required duration of action; furthermore, for a caesarean delivery, a T4 level of sensory block is necessary. It has been shown that hyperbaric bupivacaine provides effective anesthesia for caesarean deliveries with minimal motor block and a low incidence of intraoperative complication [16].

As showed by Dobrydnjov et al., intrathecal α2 adrenergic agonists use prolonged analgesia. This system is the result of both presynaptic (a decrease in substance P release from primary afferent nerve terminals) and postsynaptic (an increase in the hyperpolarization of nerve terminations) circumstances of intrathecal α2 adrenergic agonists [17]. In pregnant women, clonidine appears to have a synergistic antinociceptive effect with endogenous opioids occurring during late pregnancy. Clonidine also appears to enhance the effects of epidural and intrathecal opioids. Sedation levels highly increase in accordance with the clonidine dose. This is known to be caused by the activation of alpha-2 adrenergic receptors in the central nervous system (CNS) that reduce the release of noradrenaline from the locus coeruleus, which in turn increases inhibitory interneuronal gamma-amino-butyric acid (GABA) activity and leads to sedation and anxiolysis [18]. The outcome is dose-dependent, independent of the route of administration, and it has an effect that begins 20 to 30 min post-injection [19].

Dexmedetomidine is a highly selective adjuvant used in anesthesia. There are several reliable mechanisms to explain the anesthetic ability enhanced by the α2 adrenergic agonist. According to some scientists, the action of the α2-agonism of dexmedetomidine induces vasoconstriction, which might help lengthen the duration of analgesia. In addition, dexmedetomidine potentiates spinal block via a synergistic interaction between α2 receptors and sodium channels, resulting in a dose reduction of the local anesthetics required for achieving effective spinal anesthesia for certain surgical procedures [8]. Dexmedetomidine 10 mcg was administrated in spinal anesthesia in order to enhance the anesthetic effect of local anesthesia. Dexmedetomidine—together with bupivacaine—can increase the duration of anesthesia with shorter sensory and motor block onsets compared to bupivacaine alone. In addition, it guarantees hemodynamic stability and reduces the use of a rescue dose of an analgesic drug [2]. Dexmedetomidine prolongs spinal anesthesia because of its supra-spinal action at the locus coeruleus and dorsal raphe nucleus. In addition, dexmedetomidine is a more selective α2-agonist receptor than clonidine, with greater sedative and analgesic effects. Furthermore, the outcomes show that the addition of clonidine, an α2-agonist, to hyperbaric bupivacaine extends both the duration of effective analgesia and the sensory and motor blocks compared to the bupivacaine–fentanyl combination. This result is similar to those of previous studies, which reported an incidence of shivering of 10–30% in the control groups and no shivering in the dexmedetomidine ones. Intrathecal dexmedetomidine highly reduced the occurrence of shivering in the caesarean deliveries; it inhibited the body’s thermoregulatory center because it blocked the transmission of body temperature information at the level of the spinal cord. This was important because shivering increased oxygen consumption and carbon dioxide production, which had an impact on maternal physiological functions. Dexmedetomidine can decrease the necessity of opioid use during a caesarean delivery, thereby reducing the risk of nausea and vomiting. In addition, in the postoperative period, it had an opioid-sparing effect, reducing the rescue dose of opioids and analgesics [3]. During a caesarean delivery, intrathecal dexmedetomidine appears to be safe both for the mother and fetus. Dexmedetomidine is not easily transferred through the placenta due to its fat-soluble characteristics, so it does not increase neonatal adverse reactions during a caesarean delivery. No important differences in the neonatal 1 and 5 min Apgar scores or cord blood gas parameters were observed. Furthermore, Dexmedetomidine did not raise the incidence of adverse reactions such as maternal bradycardia and hypotension [15]. No abnormal symptoms or signs in the nervous system were found, which suggests that dexmedetomidine is a safe intrathecal adjuvant.

When sufentanil is added to hyperbaric bupivacaine in the subarachnoid space, the duration of analgesia is highly extended compared to that of bupivacaine alone. Nevertheless, increasing the dose of intrathecal sufentanil above 10 µg does not achieve an equivalent increase in the duration analgesia [20]. The incidence of itching or pruritus markedly varies in the literature from 30% to 80%, or even up to 95% in the post-operative period. Some studies showed pruritus’ incidence and gravity are proportional to the opioid dose [12]. The addition of sufentanil, at less than 10 µg, to intrathecal hyperbaric bupivacaine has been recently administered in a caesarean delivery [13]. Wilwerth et al., according to their results, state that sufentanil 5 µg might be the dose of choice as it seemed to be associated with the best quality of anesthesia without any increase in side effects [1]. Sufentanil 2.5 µg, or a greater dose, increases the incidence of pruritus in a very significant way. Together with pruritus, opioids delivered by the spinal route may cause nausea, vomiting, urinary retention, and respiratory depression, mainly owing to opioid action via µ and κ receptors [21]. The combination of Sufentanil and hyperbaric bupivacaine was administered at different intrathecal doses, and pruritus resulted in being the most common side effect, almost always being connected to high doses of sufentanil [22]. The incidence of pruritus was reported to range from 20% to 80% at different doses of intrathecal sufentanil [22,23,24]. Both adjuvants were also shown to be safe in pregnant women with SARS-COV-2 infection undergoing caesarean delivery [24,25].

The incidence of hypotension for both groups was less than 30%; this is the reason for the approach taken by boli regarding the use of an inotropic drug. All the patients that reported this side effect had aortocaval compression syndrome listed in their medical history. This syndrome is even defined as a supine hypotensive syndrome. This pathophysiologic state occurs in pregnant women, mainly after 20 weeks gestation, and in the supine position. The inability of blood to flow back from the lower extremities to the maternal heart and central circulation occurs via the uterus’s compression of the inferior vena cava and aorta. As a consequence, this limits blood flow to the placenta and may cause morbidity and mortality both for the mother and fetus [26,27].

The intrathecal use of sufentanil as an adjuvant is a standard treatment in our institute, and it is rarely employed with other intrathecal adjuvants such as clonidine or dexmedetomidine in the same anesthetic mixture. As a rule, a combined spinal epidural (CSE) is our choice of anesthetic when the surgical time is probably longer or when postoperative pain can be higher than in a caesarean delivery (for example, due to prior abdominal surgeries). Dexmedetomidine is used as alone as an adjuvant in spinal anesthesia in pregnant women with postoperative shivering in their medical history or when a complication regarding intrathecal opioids is reported.

Our study has some limitations. Firstly, the retrospective framework of the study posed a risk of bias based on the analysis of the results; secondly, the clinicians selected patients for adjuvant administration based on their medical history and not all patients were cared for by the same intraoperative and postoperative team; thirdly, our study involved a relatively small number of patients; fourthly, the neonatal outcome data included solely the Apgar score because an umbilical blood gas analysis was not present for all patients; and finally, many other factors can have a significant impact on pain, nausea, vomiting, and even shivering. Furthermore, some possible confounders are anxiety and pain catastrophizing. Therefore, we advocate that our discoveries will be further validated or contradicted by future studies with a randomized prospective design in order to modify some elements in this area of greatest significance.

## 5. Conclusions

We have concluded that adding dexmedetomidine (10 µg) and sufentanil (5 µg) as adjuvants to hyperbaric bupivacaine produced an adequate degree of anesthesia and postoperative analgesia. Dexmedetomidine does not cause superior or equal analgesia compared to sufentanil, unlike what can be observed in the literature. In the dexmedetomidine group, a VAS score of six in the first 24 h was the worst score reported, while in the sufentanil group a VAS score of 3 was the worst score. The degree of postoperative analgesia was better in the group administered intrathecal sufentanil but the incidence of side effects such as pruritus was higher in this groups of patients. Dexmedetomidine and sufentanil prolong the sensory and motor block of the duration of bupivacaine’s spinal anesthesia. Adjuvant dexmedetomidine prevented postoperative shivering. Dexmedetomidine as an adjuvant did not increase the incidence rate of hypotension.

Nevertheless, further studies are required before establishing dexmedetomidine or sufentanil as choice adjuvants for caesarean deliveries. Furthermore, prospective randomized studies may demonstrate, with statistically significant evidence, that chills do not occur when using dexmedetomidine and it can be considered as the intrathecal choice adjuvant in patients with a history of shivering and pruritus under previous anesthesia.

## Figures and Tables

**Table 1 jcm-11-06868-t001:** Demographic and surgical characteristics.

	Group 1 (*n* = 24)	Group 2 (*n* = 28)	*p*-Value
Age (years)	32 ± 6	31 ± 5	0.4
BMI (kg/m^2^)	26 ± 3	27 ± 6	0.4
ASA II physical status	24 (100%)	28 (100%)	1
Gestational age (mean week)	39.9 ± 1	39.2 ± 0.9	0.8
Duration of surgery (mean minute)	44 ± 2	45 ± 3	0.1
Time of onset of sensory block (min)	7.3 ± 0	7.4 ± 0	0.6
Patients with one or more previous pregnancies (number)	11 (46%)	15 (54%)	0.6
Blood loss during surgery (mean milliliters)	689	674	0.9
Incidence of Hypotension (number)	7 (29%)	8 (28%)	0.9

Values are mean, standard deviation (SD), or number of patients (proportion, %). BMI (body mass index); ASA (American Society of anesthesiologists).

**Table 2 jcm-11-06868-t002:** Indications for elective cesarean section.

	Group 1 (*n* = 24)	Group 2 (*n* = 28)	*p*-Value
Previous cesarean section (number)	8 (33%)	10 (36%)	0.9
Maternal pelvic deformity (number)	2 (8%)	2 (7%)	0.9
Abnormal fetal presentation (number)	5 (21%)	6 (21%)	1
Disproportion in size between fetus and maternal pelvis (number)	2 (8%)	3 (11%)	0.8
Maternal request (number)	5 (21%)	4 (14%)	0.5
Fetal pathology (number)	2 (8%)	3 (11%)	0.8

Values are given as mean, standard deviation (SD), or number of patients (proportion, %).

**Table 3 jcm-11-06868-t003:** Outcomes.

	Group 1 (*n* = 24)				Group 2 (*n* = 28)				*p*-Value
	1	2	3	4	1	2	3	4	
Bromage scale	24 (100%)	0 (0%)	0 (0%)	0 (0%)	28 (100%)	0 (0%)	0 (0%)	0 (0%)	*
Hollmen scale									
T3	7 (29%)				8 (28%)				0.9
T4	15 (63%)				19 (68%)				0.5
T5	2 (8%)				1 (4%)				0.5
VAS h24 after surgery	4 ± 2				2 ± 1				0.01
Shivering	0 (0%)				2 (7%)				0.2
First flatus time									
within 12 h	24 (100%)				7 (25%)				<0.05
between 12–24 h	0 (0%)				20 (39%)				<0.05
Nausea	1 (4%)				4 (14%)				0.2
Vomiting	1 (4%)				4 (14%)				0.2
Itching	0 (0%)				10 (36%)				0.01
Apgar 1 min									
<7	0 (0%)				0 (0%)				*
>7	24 (100%)				28 (100%)				
Apgar 5 min									
<7	0 (0%)				0 (0%)				*
>7	24 (100%)				28 (100%)				
Motor recovery time	128 ± 2				127 ± 2				0.1

Values are given as mean, standard deviation (SD), or number of patients (proportion, %). * constant.

## Data Availability

Not applicable.

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
