# Peer review of "Sufentanil vs. Dexmedetomidine as Neuraxial Adjuvants in Cesarean Section: A Mono-Centric Retrospective Comparative Study"

_jcm, 2022, doi:10.3390/jcm11226868_

Round 1
Reviewer 1 Report
General comments: I read the manuscript with interest and here is a list of major concerns that the authors may want to address:
1) In the Methods section, I am not sure to include women aged between 1 and 45 years is appropriate. This is because women before menarche cannot have pregnancies and, in addition there should be a reason why women older than 45 years old were excluded
2) Again in the Methods section: inclusion criteria is term pregnancy, but then five women with gestational age between 36 and 40 weeks were excluded.
3) Why were preterm pregnancies excluded? Is there any anesthesiologic implication due to gestational age? If yes, this should be adequately addressed.
4) There are no information regarding pregnancy in terms of both previous pregnancies and the index pregnancy. This information is important since it can have an influence on anesthetic procedures (if a woman has preeclampsia her hemodynamic status may be affected by the baseline condition). In addition, it would be important to know why women underwent cesarean delivery.
5) Also demographic characteristics should be implemented. Including only age, BMI and ASA status is really poor. For example parity is important in women recovery after cesarean section so this is one of the data that should be included.
To add data is important to be sure that what the Authors observed is due to the different adjuvant employed and not to other variables that were just not included
6) A paragraph of clinical definitions should be added to the Methods section. This manuscript would be important not only to the Anesthesiologists but also the Gynecologists so it should provide more data and more definitions in order to widen the audience
Reviewer 2 Report
the authors apparently pursued a retrospective study of outcomes associated with spinal anest with bup 10mg and either sufent 5mcg or dex 10mcg among 52 otherwise healthy asa physical status 1-2 parturients undergoing cesarean delivery. they found no differences in limited demographic data between groups and minor differences in pain at 24h (lower in bup + suf group) and shivering and itching (when and how were these outcomes assessed?) (less shivering and itching in bup + dex group). the manuscript is interesting for exploring associations with IT dex but there are numerous flaws in description and quite possibly study design that may poison any meaningful conclusion.
major comments:
1. abstract is poor. abstract should state in clear terms study design. no discussion of hypothesis, primary outcome, other outcomes of interest, study population (aside from the institution - which is meaningless to most readers). i had no idea this was a retrospective study until much later in the manuscript. the abstract misrepresents the study and overstates the conclusion. i had no idea of the population size, no idea of odds of whatever outcome they were actually interested in occurring in whichever group until two pages into the manuscript. this should be in the abstract. later in the abstract the authors state dex has shorter onset than bupiv alone but the limited study description stated the design was comparing bup with dex or bup with sufent, not bup alone. that makes absolutely no sense. was it three groups (bup with dex, bup with suf and bup alone) or did they mean bup + dex was "faster" than bup + suf? in any case, unclear, bad. further, in the results section, there are no outcomes reported of time to incision or sensory block of this or that. this conclusion has no results in this manuscript that supports it.
2. pg2 ln 50-53: the authors state that alpha2 agonists do not induce hypotension and bradycardia and do not affect neonatal outcome by siting two studies, one of which is a small meta-analysis. much literature exists to state that a2 agonist do in fact contribute to hypotension and bradycardia and sedation such as: Filos KS et al. hemodynamic and analgesic profile after intrathecal clonidine in humans. a dose-response study. Anesth. 1994;81(3):591-601, Chiari a, et al. analgesic and hemodynamic effects of IT clonidine as the sole analgesic agent during 1st stage of labor. Anesthesiology. 1999; 91(2):388-96, Missant C, et al. IT clonidine prolongs labour analgesia but worsens fetal outcome. Can J Anaesth. 2004;51(7):696-701, Belhadj Amor M, et al. 30 mcg IT clonidine prolongs labour analgesia, but increases the incidence of hypotension and abnormal foetal heart rate patterns. Ann Fr Anesth Reanim. 2007;26(11):916-20 and more.
3. pg 3 ln 101-103: ". . . Bupivacaine 0.5% 10 mg and dexmedetomidine 10 μg (3.5 milliliter) or Bupivacaine 0.5% 10 mg and Sufentanil 5 μg (3.5 milliliter) was injected in according to the group." this is a retrospective study. this implies a randomized controlled study. in a retrospective study, patients received a treatment and the investigators look for an associated outcome. at best, if this really happened, it should be stated something like the institutional standard spinal anesthesia consisted of bupivicaine 10mg (hyperbaric?) with either sufent 5mcg or dex 10mcg or we selected patients who received bup 10mg and either dex or suf.
4. no comments on surgical conditions such as number of prior cesareans or blood loss. these could be important variables that affect outcomes such as pain and should be readily available and described in the demographics btw groups. were these all scheduled cesareans? were any intrapartum?
5. what, if any, known factors led clinicians to use dex or suf in addition to bup for spinal anesthesia? did the clinicians select dex when the patients had multiple prior abd surgeries or cesareans and suspected prolonged surgery? why did clinicians use dex or suf?
6. pg 3 ln 121: what is recanalization, let alone how is 'recanalization' assessed? recanalization does not translate into english for it to be reported as an outcome, i need to know what it is and how it was assessed.
minor comments: grammatical errors: pg 1 ln17: would be better if reworded to, ". . . complete and predictable nerve block with fast onset and few complications." pg 1 ln 26 should be shivering rather than shiver. throughout the manuscript, please use a consistent spelling of caesarean, at times the authors use cesarean and others they use caesarean. also, please do not use caesarean section, rather use caesarean delivery throughout. pg 1 ln 35: block not blockage. pg 1 ln 37: "Though spinal anesthesia presents adverse reactions that still seriously threaten the safety of maternal and fetal life." incomplete sentence. pg 1 ln 38: shivering not shiver. pg 1 ln 41-2: should be "Several studies have proven that by reducing the dose of intrathecal LA, the incidence of adverse events is reduced." pg3 ln94: 2.5, not "2,5". there are other english grammatical errors scattered through the remaining manuscript that could and should be corrected. pg 3 ln 106-107: define tachycardia and bradycardia. pg 5 ln 171: sensory not 'sensitive'.Author Response
Please see the attachment.

Round 2
Reviewer 1 Report
I would like to compliment with the Authors for the efforts provided in addressing my comments.
The manuscript is, in my opinion, suitable for publication and can be accepted in its current form.
Best regards
Salvatore Andrea Mastrolia
Reviewer 2 Report
1. abstract is improved but should still be improved further. for example, use of english in the abstract and elsewhere needs to be improved. multiple times the term shiver and/or itch is used when it should be shivering and/or itching.
2. additional discussion of a2 agonist addressed.
3. there remains no discussion of why clinicians selected dex or suf.
4. no tables are included in the manuscript i have here. nothing appears between section 3.6 and 4. it is entirely blank. even the tables from the first version of this manuscript are not present. i still do not know the indications for CD, the blood loss, OR time, the parity, or other important surgical confounders btw subject groups. fluid replacement, total dose of phenylepherine and ephedrine, episodes of hypotension such as time below a certain MAP are not present. without these factors, i would recommend rejection of this manuscript.
5. i still don't know how shivering, itching, nausea and vomiting was assessed. did the clinician ask the patient? if so when? any complaint of these conditions? intraop or postop? anytime in the first 24h postop? was it clinician judgement? where these variables dichotomous or continuous variables as in a likert scale? this is one of their seemingly primary outcomes. were rescue antiemetics used? prophylactic antiemetics (ondansetron)?
6. while not a deal breaker, it is antiquated to not use a prophylactic infusion of phenylepherine after spinal anesthesia for CD as opposed to intermittent boluses.
7. section 2.6 is redundant and entirely unnecessary. all of this section should either be in an appendix or simply references to other articles.
8. limitations section needs to be far more explicit. you have presented no indication there wasn't bias in clinicians selecting patients for this or that treatment. your previous response to this reviewer regarding selection of patients remains inadequate. you stated (and did not put anything in the manuscript in this regard) that clinicians selected dex if pt has some indication they had itching previously. what? what scale? was it elicited history (clinicians asked) or recorded in the medical record or what? none of this was entered in the manuscript.
10. what was the surgical and anesthetic team? of course not all patients were cared for by the same intraop and postop team. at the least this should be mentioned in the limitations.
this is a reasonable question to ask. however, the completeness and presentation of data supporting the conclusions is entirely inadequate. so many other things can have significant impact on pain, nausea, vomiting, even shivering are not at all addressed here. chief among these possible confounders is anxiety and pain catastrophizing. at a minimum, a more complete presentation of data such as inclusion of surgical factors such as prior abdominal surgeries, blood loss, OR time, preoperative factors such as preoperative opioid use, anxiety, depression, intraoperative hemodynamics including dose of pressors and time under a certain MAP (such as >20% below baseline or MAP <60) need to be included or discussed in a much expanded limitations section. with regards to pain, the authors mention VAS assessment. when? average of all scores in the first 24h? worst VAS in 24h? what does VAS mean without these other considerations?
